

# Attaching artificial Achilles and tibialis cranialis tendons to bone using suture anchors in a rabbit model: assessment of outcomes

Obinna P. Fidelis[1], Caleb Stubbs[1], Katrina L. Easton[1], Caroline Billings[2], Alisha P. Pedersen[2], David E. Anderson[2] and Dustin L. Crouch[1]

[1] College of Engineering, University of Tennessee - Knoxville, TN, United States of America
[2] College of Veterinary Medicine, University of Tennessee - Knoxville, TN, United States of America

Corresponding author
Obinna P. Fidelis,
ofidelis@vols.utk.edu

## ABSTRACT

**Objective**. The purpose of this study was to investigate the timing and mode of failure of metallic screw-type suture anchors used to attach artificial tendons to bone in an *in vivo* New Zealand White rabbit model.

**Study Design**. Metal suture anchors with braided composite sutures of varying sizes (United States Pharmacopeia (USP) size 1, 2, or 5) were used to secure artificial tendons replacing both the Achilles and tibialis cranialis tendons in 12 female New Zealand White rabbits. Artificial tendons were implanted either at the time of (immediate replacement, $n = 8$), or four/five weeks after (delayed replacement, $n = 4$) resection of the biological tendon. Hindlimb radiographs of the rabbits were obtained immediately after surgery and approximately every other week until the study endpoint (16 weeks post-surgery).

**Results**. All suture anchors used for the tibialis cranialis artificial tendons remained secure and did not fail during the study. The suture anchor used to attach the Achilles artificial tendon to the calcaneus bone failed in nine of 12 rabbits. In all cases of suture anchor failure, the suture broke away from the knot, while the metallic screw remained securely embedded in the bone. Based on radiographic analysis, the mean estimated failure timepoint was $5.3 \pm 2.3$ weeks post-surgery, with a range of 2–10 weeks. Statistical analyses (Mann–Whitney $U$ test and Fisher's exact test) revealed no significant effect of tendon implantation timing or suture size on either the timing or frequency of suture anchor failure.

**Conclusion**. For the suture anchors used to attach artificial tendons in this study, suture anchor failure was most likely due to suture wear or cutting against the eyelet of the anchor screw. Future studies are needed to test the effect of suture-eyelet interaction on suture strength under different loading conditions.

## INTRODUCTION

Portions of this text were previously published as part of a preprint (https://www.biorxiv.org/content/10.1101/2024.04.29.591695v1).

A suture anchor consists of a suture and an anchor with an eyelet on the anchor through which the suture passes. Suture anchors are frequently used in human and veterinary musculoskeletal reconstruction to reconnect soft tissues, such as tendons and ligaments, to bones (*Cho, Bae & Kim, 2021*; *Visscher et al., 2019*). Suture anchors can offer quick, reliable, and stable fixation (*Burnham et al., 2020*; *Ravin, Gonyon & Levin, 2005*), and can achieve close apposition between the soft tissue and bone (*Altiparmak & Uckan, 2013*). An intimate interface between the anchor and suture is needed to ensure that the soft tissue is securely connected to the bone to facilitate healing (*McFarland et al., 2005*). Compared to other techniques for fixation of soft tissues to bone, the strength of suture anchors is superior to other means of fixation such as bone tunnels, staples, screws, washers, or tapered plugs (*Barber, Cawley & Prudich, 1993*).

Anchors can be classified as either screw-type or non-screw-type (*Barber, Herbert & Click, 1995*), with screw-type anchors having greater load-to-failure resistance compared with non-screw-type designs (*Barber et al., 2006*; *Barber, Herbert & Richards, 2003*). The eyelets in screw-type anchors can be either raised above the screw or embedded within it. Anchors may also be bioabsorbable, nonmetallic, or metallic (*Barber, Cawley & Prudich, 1993*). Metallic anchors are relatively affordable with minimal undesirable biological reactions, ensuring safe and long-term fixation of the anchor within the bone while the tissue heals (*Longo et al., 2019*).

Mechanical failure of a suture anchor is a potential complication for clinical concern as it compromises the stability and integrity of the surgical reconstruction (*Meyer et al., 2002*). When failure occurs, the connection between soft tissue and bone becomes weak or ineffective, resulting in the need for revisions. Revision surgeries are often complex, costly, and burdensome for the patient. Sometimes, revisions may lead to more loss of bone tissue, depending on the nature of the reconstruction, as the bone is prepared to hold the anchor (*Panegrossi et al., 2014*). This highlights the importance of developing reliable and durable suture anchors to minimize the risk of failure. Several factors may affect the durability of suture anchors, including suture anchor design, material composition, and surgical techniques (*Fleischli, 2018*; *Schanda, Obermayer-Pietsch & Sommer, 2022*).

Sutures associated with suture anchors tend to fail in one of two ways: at the knot, which is considered the weakest point of a tied suture, or at the eyelet of the anchor (*Meyer et al., 2002*). Using a model of bovine infraspinatus tendon repair, suture breakage at the eyelet was the most common mode of failure for metallic suture anchors with eyelets raised above the screw, when the anchor head was flush with the surface of the bone, and when the anchor head was raised above the bone (*Bynum et al., 2005*). Suture breakage also was the predominant mode of failure for two metallic screw type suture anchors evaluated in a human cadaveric shoulder using an arthroscopic Bankart method, although the study did not state if the sutures failed at the knot or around the eyelet (*Barber, Cawley & Prudich, 1993*). In an extensive study of the mechanical properties of suture-anchor interactions, fourteen different sutures were used in combination with thirty-one anchors; suture breakage at the midpoint of the tested strands was the failure mode for all the suture anchors (*Barber et al., 2006*; *Barber, Herbert & Richards, 2003*).

In a previous study, a metallic screw-type anchor was used to attach artificial tendons in rabbits. The suture failed away from the knot in multiple rabbits, prematurely. Mechanical testing of the anchor using a size 5 suture yielded a failure load of $437.24 \pm 42.35$ N with cycling and $498.13 \pm 18.20$ N without cycling (*Fidelis et al., 2024*). The objective of our study was to investigate the timing and mode of failure of metallic screw-type suture anchors used to attach artificial tendons to bone in an *in vivo* New Zealand White rabbit model. We hypothesized that suture size and timing of tendon replacement surgery would have a significant effect on the timing and frequency of suture anchor failure. The results of our study will inform the design and selection of suture anchors for clinical applications.

## MATERIALS AND METHODS

### Artificial tendon fabrication

We fabricated custom polyester suture-based silicone-coated artificial tendons to replace the biological Achilles and tibialis cranialis tendons, as previously described (*Hall et al., 2023*). The artificial tendon design was adapted from an existing device (*Melvin et al., 2010*; *Melvin et al., 2012*) using customized United States Pharmacopeia (USP) size 0 braided polyester sutures (RK Manufacturing Corp, Danbury, CT, USA). The sutures were cut to a length of 12 inches, double-armed with swaged 3/8-circle tapered point needles (0.028-inch wire diameter). Artificial tibialis cranialis tendons included two suture strands, folded in half to form a loop, then braided to the desired length yielding four suture threads for muscle attachment; Artificial Achilles tendons were made using three suture strands that were folded in half to form a loop at the mid-point, then braided to the desired length, yielding 6 suture threads for muscle attachment (Fig. 1). The sutures were secured around a cylindrical pipe to form the looped end of the tendon at the start of braiding. The length of the braided section was estimated based on the length of the biological tendon, as determined from pre-surgery hindlimb radiographs. To prevent tissue adhesion, medical-grade biocompatible silicone (BIO LSR M340; Elkem Silicones, Lyon, FR) was applied to the braided portion of the tendon. Two parts (ratio 1:1) of the silicone was manually mixed by stirring for up to 5 min in a cup. The mixture was transferred into a syringe and centrifuged at 4,900 rpm for 4 min to eliminate air bubbles and achieve a homogenous mixture. The mixture was then applied to the braided section of the artificial tendon to create a thin and uniform silicone coating on the braided suture. The tendons were cleaned in an ultrasonic cleaner (Model: JPS-08A) and sterilized using hydrogen peroxide gas to prepare them for surgery. We applied a silicone coating on the braided polyester sutures to deter tissue adhesion and encourage the formation of a pseudo-sheath to allow the tendon to slide relative to surrounding tissues (*Melvin et al., 2011*).

### *In vivo* study

All animal procedures were prepared, registered and approved by the Institutional Animal Care and Use Committee at the University of Tennessee, Knoxville (Protocol #2726). The study included twelve female New Zealand White rabbits (Robinson Services Inc, USA), with an average age of $19.9 \pm 2.17$ weeks and an average body mass of $3.52 \pm 0.35$ kg at the time of first surgery. All rabbits were part of a larger study to determine the effect of

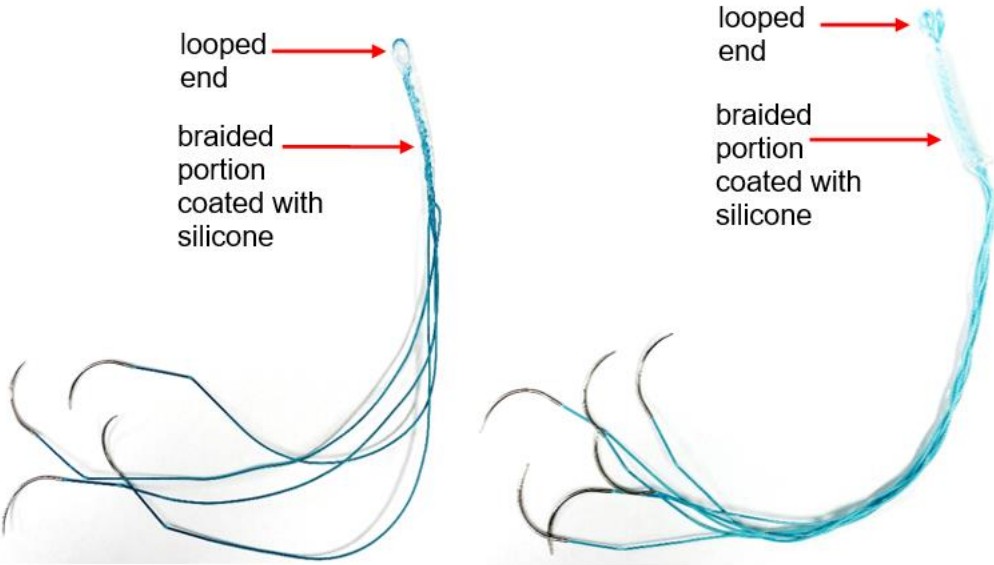

**Figure 1** **Sample artificial tendons implanted in this study.** Left: artificial tibialis cranialis tendon. Right: artificial Achilles tendon.

tendon replacement timing, immediate *versus* delayed (5 weeks), on hindlimb movement biomechanics. We housed the rabbits individually and allowed them to acclimate for a minimum of 2 weeks prior to surgery. They received daily positive human interaction and enrichment in addition to being fed an ad libitum regular laboratory diet, Timothy hay, and greens. In addition, rabbits were given playpen time twice a week for at least ten minutes before surgery and beginning two weeks after the procedure, when the bandages had been taken off. Surgeries were performed iteratively, on multiple days over a 35-week period (Fig. 2) under general anesthesia and the rabbits were given pain management with bandaging of the operated limb, post-surgery. The rabbits were administered hydromorphone (0.1 mg/kg IM) as an analgesic and midazolam (1 mg/kg IM) as sedative prior to surgery; with additional dose of hydromorphone (0.1 mg/Kg IM) post-surgery if necessary, as part of pain management. We applied silver sulfadiazine to the incision and the operated limb was bandaged for at least three days with splint bandage. Isoflurane was administered during surgery *via* face mask to induce general anesthesia. To maintain general anesthesia, isoflurane gas was evaporated in 100% oxygen.

First, the biological Achilles and tibialis cranialis tendons were excised from the musculotendinous junction to the tendon enthesis in the left hindlimb. To reduce the number of animals, no control group was used. Artificial Achilles and tibialis cranialis tendons were then implanted in the left hindlimb, either at the time of (immediate, $n = 8$) or 5 weeks after (delayed, $n = 4$) biological tendon excision (Table 1). The proximal ends of the artificial tendons were sewn into the distal end of the respective muscle. The tibialis cranialis tendon implant had four strands of suture material with swaged-on needles. The sutures were folded at their mid-point, to create the distal loop used for attachment to

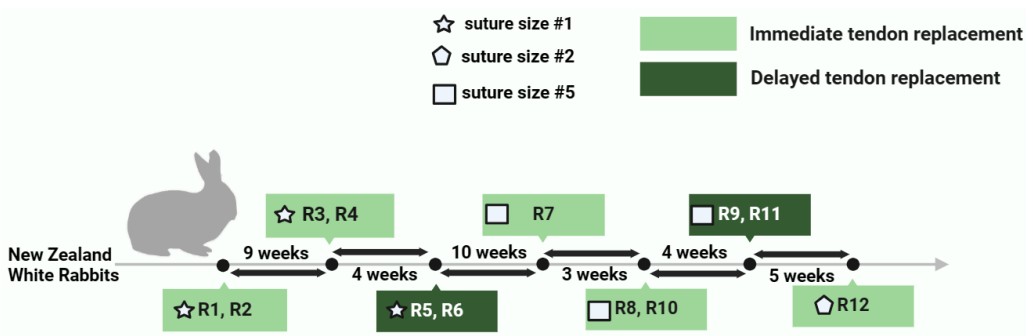

**Figure 2** Timeline of surgeries in the study showing the tendon replacement timing (immediate or delayed) and the suture size used for the Achilles tendon suture anchor for rabbits R1 through R12.

the bone anchor. The free-ends with the swaged-on needles remained unbraided and were used to attach the tendon implant to the muscle. Muscle attachment was established using one-pair of free sutures to create two pairs of modified single-locking loops in the muscle and tying these two sutures together (*Rawson, Cartmell & Wong, 2013*). This was repeated for the second pair of suture free ends. The artificial tendon implant was attached to the muscles of the Achilles tendon in similar fashion, except that these tendon implants were composed of 6 strands of sutures resulting in three pairs of locking loops placed in the muscles. The distal ends of the artificial tendons were attached near the respective biological tendon insertion points on the bone using metallic screw-type anchors with eyelets located within a raised post. The knotted ends of the sutures on the anchors were placed at the distal incision, near the 2nd metatarsal bone for the tibialis cranialis tendon and near the calcaneus for the Achilles tendon. The skin around all incisions was then closed with an intradermal pattern using 3-0 PDS.

Two bone anchors, manufactured for use in veterinary surgery and commercially available, were chosen based on their size and ability to select suture sizes to be used. The anchor (Fig. 3) for the artificial Achilles tendon (part no. 60-27-09, 2.7 mm × 9 mm, IMEX Inc., Longview, TX) was inserted in a proximal to distal angle through the cranial aspect of the calcaneus close to the point of insertion of the biological tendon. Either USP size 1 ($n = 6$), 2 ($n = 1$) or 5 ($n = 5$) braided composite suture (Fiberwire, Arthrex Inc., Naples, FL) was passed through the eyelet for use to secure the artificial tendon to the anchor. The anchor for the tibialis cranialis artificial tendon (2 mm × 6 mm; Jorgensen Laboratories LLC, Loveland, CO, USA) was inserted in the proximal, dorsal aspect of the 2nd metatarsus (MT II) at the point of insertion of the tendon and accompanied with either a size #1 or #2 suture.

Radiographs of the operated limbs were taken at the time of surgery and approximately once every other week post-surgery to monitor the placement and integrity of the suture anchors and artificial tendons. Radiographs of both the lateral and cranial-caudal views were taken at each timepoint. Potential confounders such as order of treatment and measurements were not controlled. All authors were aware of the rabbit group allocations at all stages of the study.

**Table 1 Summary of results for Achilles artificial tendon suture anchors.**

| Rabbit ID | Study Group based on tendon replacement timing (Immediate or delayed) | USP suture size for Achilles tendon suture anchors | Postoperative radiographic timepoint at which suture anchor failure was first suspected (weeks post-surgery) | Location of suture breakage | Gap length on the day of surgery (mm) | Gap length at suspected failure timepoint or end of study (mm) |
|---|---|---|---|---|---|---|
| R1 | Immediate | 1 | 2 weeks | Mid-section | 0.000 | 6.312 |
| R2 | Immediate | 1 | 4 weeks | Mid-section | 0.720 | 6.312 |
| R3 | Immediate | 1 | No failure | – | 1.760 | 2.410 |
| *R4 | Immediate | 1 | No failure | – | 4.827 | 4.442 |
| R5 | Delayed | 1 | 6 weeks | Mid-section | 0.673 | 13.179 |
| R6 | Delayed | 1 | 4 weeks | Mid-section | 0.821 | 3.612 |
| R7 | Immediate | 5 | No failure | – | 0.564 | 0.991 |
| R8 | Immediate | 5 | 7 weeks | Mid-section | 0.344 | 4.463 |
| R9 | Delayed | 5 | 5 weeks | Mid-section | 1.057 | 12.932 |
| R10 | Immediate | 5 | 6 weeks | Mid-section | 0.815 | 4.784 |
| R11 | Delayed | 5 | 10 weeks | Mid-section | 0.798 | 5.466 |
| #R12 | Immediate | 2 | 4 weeks | unknown | 1.023 | 5.126 |

**Notes.**

*The tendon of the superficial flexor digitorum was replaced with an artificial tendon instead of the Achilles tendon, as such, this rabbit was excluded in the statistical analysis.

#This rabbit was also excluded in the statistical analysis because it was the only rabbit with a suture anchor having a size 2 suture.

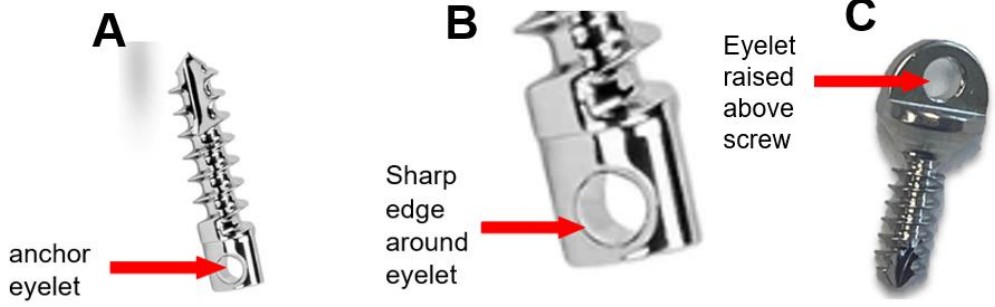

Figure 3 (A) IMEX 60-27-09 anchor (2.7 mm × 9 mm) used to attach artificial Achilles' tendon. (B) A close up image of the eyelet of the IMEX anchor showing the sharp edges around the eyelet. (C) A second raised eyelet design (JorVet 2 mm × 6 mm) used to attach artificial tibialis cranialis tendon is also shown.

At the end of the study (16 weeks post-surgery), the rabbits were humanely euthanized by intravenous overdose of pentobarbital (390 mg/ml, minimum 1 ml/10 lbs). The hindlimbs were collected *via* disarticulation of the limb at the level of the hip joint, fixed in 10% phosphate-buffered formalin for at least 5 days, and then stored in 70% ethanol. Following fixation, the hindlimbs were dissected to permit visual inspection of the suture anchor and confirmation of suture breakage in all rabbits. We also determined the mode of suture failure, either at the knot or at the mid-section (*i.e.,* away from the knot).

## Data analysis

The artificial tendon and metallic anchor were visible on radiographs, but not the connecting suture. Thus, failure of the tendon suture anchors by suture breakage was suspected when, qualitatively, the gap length between the artificial tendon and anchor was substantially longer than that observed immediately after surgery. From the digital radiographs, we used image processing software (ImageJ, NIH) to measure the tendon-anchor gap length, defined as the length from the head of anchor to the most distal visible point on the artificial tendon. No tibialis cranialis suture anchor failure occurred; thus, gap lengths were only measured for the Achilles tendon suture anchors. Post-surgery gap lengths were compared to those at the time of surgery to estimate the timing of suture anchor failure, defined as the first radiographic timepoint at which gap length had increased substantially (>five mm) compared to the previous timepoint with no return to the previous shorter gap length.

The estimated time of suture failure for the Achilles tendon suture anchor was used to perform a survival analysis to determine the probability of failure as a function of time post-surgery. For all samples, we defined suture failure as the timepoint at which a large gap distance was detected radiographically. The exact time of failure could not be determined because of the pre-determined radiography timepoints. The number of events, cumulative survival, and cumulative failure (1 −cumulative survival) was calculated for each time point using Eq. (1).

$$cumulative\ survival = \frac{survivals\ at\ time\ point - number\ of\ failures\ at\ time\ point}{survivals\ at\ time\ point}. \quad (1)$$

The survival distribution function was calculated by multiplying the cumulative survival rates and plotted against time to obtain a Kaplan–Meier survival curve. We applied the fundamental assumptions of the Kaplan–Meier survival analysis (*Etikan, Abubakar & Alkassim, 2017*).

A non-parametric analysis (two-tailed Mann–Whitney U test) was used to determine the potential effect of two key variables, surgery type (immediate and delayed tendon replacement) and suture size (1 and 5), on the timing of suture anchor failure. Additionally, the possible effect of surgery type and suture size on the *frequency* of suture anchor failure was investigated using a Fisher's exact test with the data grouped as *number of failures* associated with each surgery type and with each suture size. Two rabbits were excluded from both statistical analyses: R4 in which the tendon of the superficial flexor digitorum (SFD) was unintentionally replaced instead of the Achilles tendon, and R12 which was the only rabbit with a size 2 suture. All ten remaining rabbits were included in Fisher's exact test, and only the eight remaining rabbits with failed suture anchors were included in the Mann–Whitney *U* test (Table 1). For all statistical comparisons, a *p*-value <0.05 was considered significant.

## RESULTS

Radiographs were inspected qualitatively to identify suspected suture failures. Based on this inspection, *in vivo* failure of the suture used to attach the artificial Achilles tendons

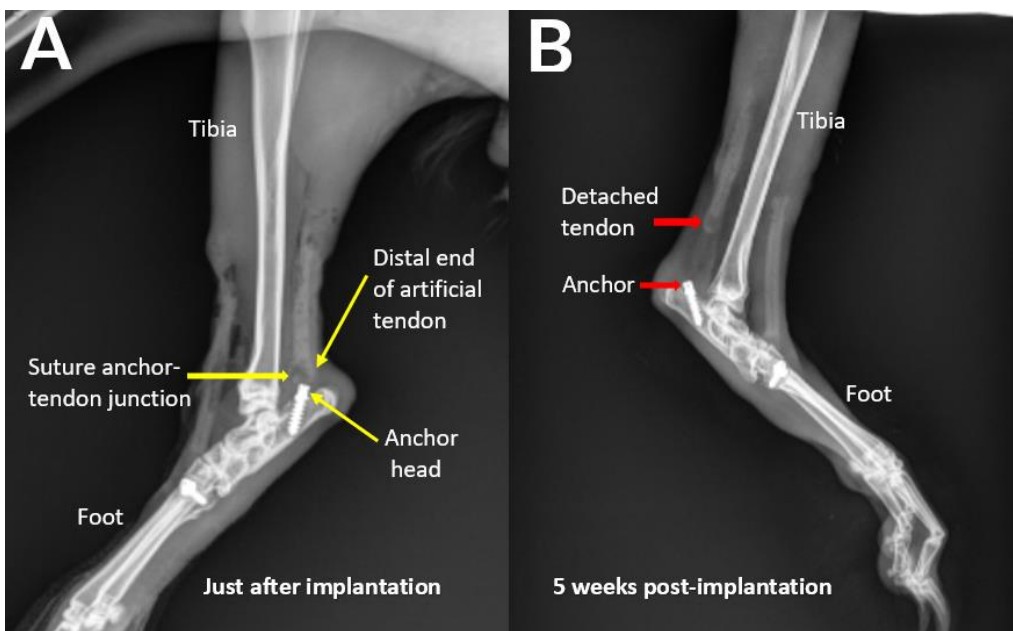

**Figure 4** (A–B) Exemplary hindlimb radiographs showing attachment of the artificial tendon just after surgery and detachment of the tendon from the anchor following failure of the suture anchor after five weeks for rabbit R9.

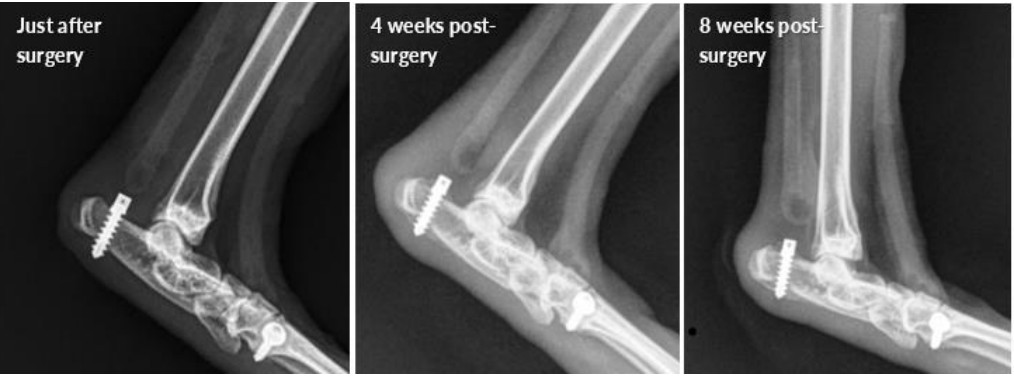

**Figure 5** Hindlimb radiographs from rabbit R11 showing a gradually increasing gap length between the anchor and Achilles artificial tendon. For the entire duration of investigation, the tibialis cranialis artificial tendon remained intact because the suture anchor used to attach this tendon did not fail, hence did not show any increase in gap length.

was initially suspected in nine of the 12 rabbits (Figs. 4 and 5). Conversely, there was no qualitative evidence of failure of the suture anchors used for attaching the tibialis cranialis artificial tendons.

All specimens were inspected post-mortem by gross dissection. For all available specimens, the failure states (failed or not) of all suture anchors determined by post-mortem dissection were consistent with those determined radiographically (Fig. 6). For all

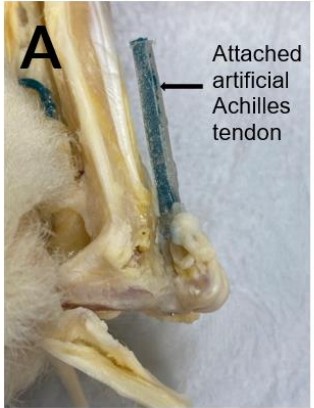
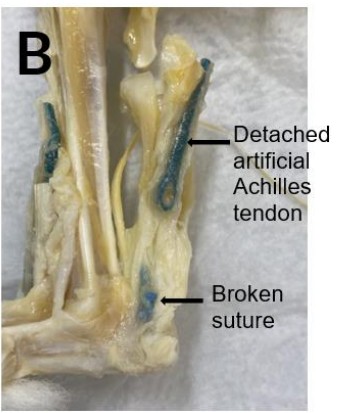
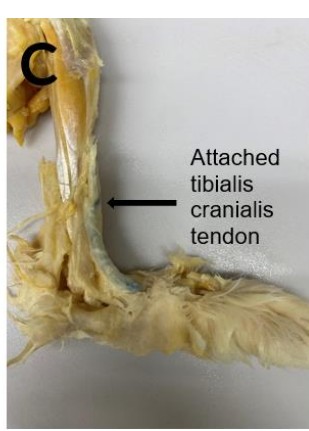

**Figure 6** (A) Artificial Achilles tendon securely attached to the calcaneus by an intact suture anchor. (B) Artificial Achilles tendon detached from a broken suture in a rabbit. (C) Artificial tibialis cranialis tendon remains attached in one of the rabbits.

failure cases, the mode of suture anchor failure was suture breakage at the mid-section, *i.e.,* away from the knot (Table 1).

The timepoint at which the substantial increase in gap length was first observed was designated as the failure timepoint. The earliest failure (R1) occurred only 2 weeks after implantation. Three sutures failed at 4 weeks post-implantation (R2, R6, and R12), one at 5 weeks (R9), two at 6 weeks (R5 and R10), and one each at 7 weeks (R8) and 10 weeks (R11) (Fig. 7). In three rabbits (R3, R4 and R7), the suture of the suture anchor did not fail. In some rabbits, gap length continued to increase over time after the suspected failure timepoint. Of the nine sutures that failed, the mean failure timepoint was 5.3 ± 2.3 weeks post-surgery, with a range of 2–10 weeks.

The Kaplan–Meier survival analysis (Table 2, Fig. 8) revealed that 92 percent of the suture anchors survived beyond the initial two weeks. As time progressed, the survival rate decreased gradually, with 67 percent of the anchors surviving beyond 4 weeks, 58 percent of the anchors surviving beyond 5 weeks, 42 percent surviving for 6 weeks, and 33 percent of the suture anchors surviving beyond seven weeks. Finally, 25 percent of the anchors survived up to the end of the study at 16 weeks. All data is presented in this manuscript and no extra data is left out.

The two-tailed Mann–Whitney $U$ test revealed no significant difference in effect of implantation timing between the delayed implantation and immediate implantation ($U = 8.0$, $p = 0.05$), with both groups having the same median score (median $= 18$). Furthermore, the test also showed no significant difference in effect of suture size between the suture anchors with suture size 1 and suture size 5 ($U = 2.5$, $p = 0.05$), with the suture size 5 group having higher median score (median $= 23.5$) than the suture size 1 group (median $= 12.5$). Similarly, Fisher's exact test indicated no significant effect of implantation timing on frequency of failure ($p = 0.071$) and no significant effect of suture size on frequency of failure ($p = 0.556$).

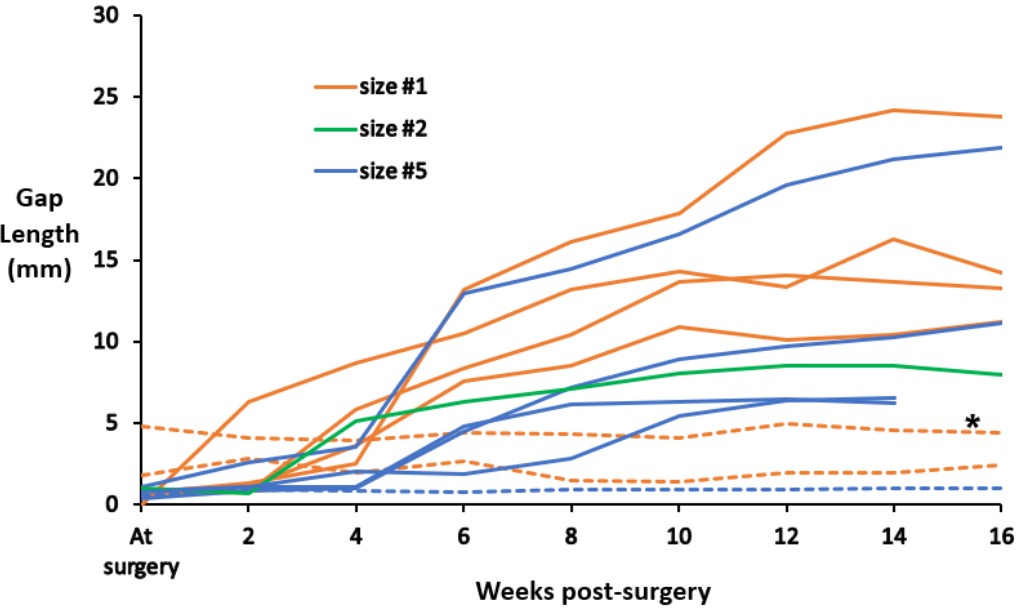

**Figure 7** Gap length between the anchor and Achilles artificial tendon as measured from radiographs. In nine rabbits, the gap length increased by at least five mm between two consecutive radiographic timepoints (solid lines); these rabbits were suspected to have suture anchor failure. The gap length remained about the same in three rabbits (dashed lines); these were suspected to have intact suture anchors throughout the entire study. *In R4, the tendon of the superficial flexor digitorum was replaced, instead of the Achilles tendon.

**Table 2  Kaplan–Meier survival rate and distribution function for 12 suture anchors.**

| Time when failure occurred (weeks) | Number of failed suture anchors | Number of censored data | Survival rate of suture anchors | Survival distribution function |
|---|---|---|---|---|
| 0 | 0 | 0 | 1.0000 | 1.0000 |
| 2 | 1 | 0 | 0.9167 | 0.9167 |
| 4 | 3 | 0 | 0.7272 | 0.6667 |
| 5 | 1 | 0 | 0.8750 | 0.5833 |
| 6 | 2 | 0 | 0.7140 | 0.4165 |
| 7 | 1 | 0 | 0.8000 | 0.3332 |
| 10 | 1 | 0 | 0.7500 | 0.2499 |
| 16 | 0 | 3 | 1.0000 | 0.2499 |

# DISCUSSION

The frequent failures of the Achilles suture anchor-tendon implant linkage were surprising and characterized by a sudden, radiographically apparent, large increase in gap length, followed by gradual continued increase in gap length. The gradual increase in gap length may have been due to shortening of the muscle fibers (*Maffulli, 1999*; *Meyer et al., 2011a*). Muscles retract after tendon ruptures, with the degree of retraction depending on the

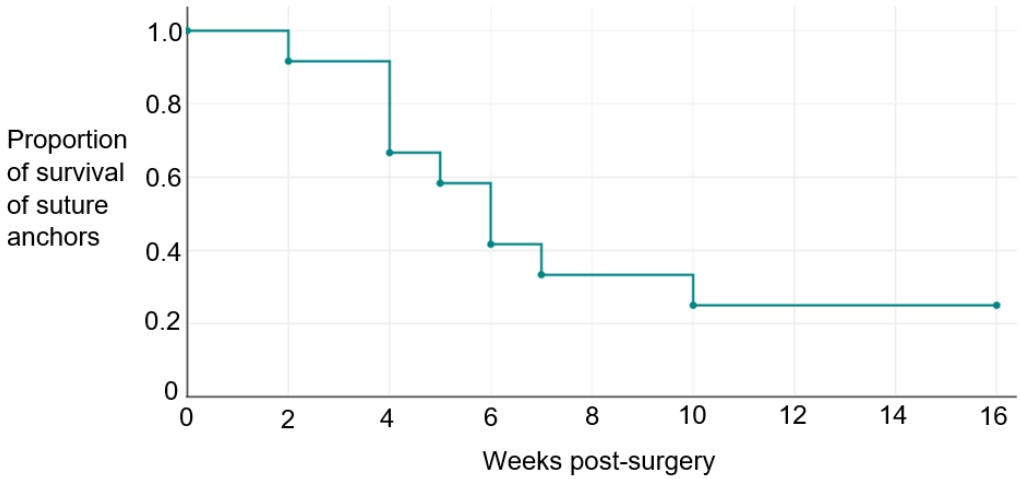

**Figure 8** Kaplan–Meier survival curve showing the probability of suture anchor failure at the respective time intervals.

muscles involved and the severity of the rupture (*Gil-Melgosa et al., 2021*; *Meyer et al., 2011b*). Mean gap length of up to 6.0 cm has been reported in cases of distal biceps tendon ruptures (*Samra et al., 2019*) and up to 4.8 cm gap length in cases of supraspinatus tendons ruptures (*Meyer et al., 2011a*). The implications of muscle retraction include muscle atrophy, loss of muscle mass and strength, muscle weakness and a loss of tissue quality and quantity, which increase the risk of complications and decrease the success rate of surgical repair (*Lakemeier et al., 2010*).

The location of suture failure (at the mid-section away from the knot in all 9 cases of failure) provides important clues about the cause of failure. The knot is considered the most severe stress point with sutures, and therefore, a weak link in a suture anchor. Therefore, failures at the knot are usually attributed to excessive loading. Conversely, suture anchor failure away from the knot suggests wear or abrasion of the suture against part of the anchor, which can reduce suture strength by as much as 73 percent (*Meyer et al., 2002*); this value corresponds with the observation that the reported *in vivo* Achilles tendon loading (57.7 ± 0.5 N) in rabbits during hopping is 80 percent lower than the strength of the size 5 braided composite suture reported by the manufacturer (295 N) (*Arthrex Inc, 2014*; *West et al., 2004*)). In our recent mechanical tests, we found that the combination of the size 5 braided composite suture with the IMEX anchor is even stronger, with a maximum strength of 437 ± 42 N after 1,000 loading cycles along the long axis of the anchor screw (*Fidelis et al., 2024*).

Suture anchor design, especially the design of the eyelet through which the suture passes, likely played a significant role in the observed failures. The eyelets of suture anchors are generally configured in two ways: either raised above the screw or embedded within the screw. Anchors with raised eyelet designs offer greater versatility because they can accommodate sutures of various sizes. However, this design has drawbacks; a raised eyelet allows relatively unrestricted freedom of suture movement through the eyelet.

Conversely, the embedded eyelet design, while limiting the range of suture sizes that can be accommodated, restricts suture motion through the eyelet. Therefore, sutures may wear more rapidly in suture anchors with raised eyelets than with embedded eyelets; future studies should quantify the effect of suture motion on the extent and rate of suture wear. For long-term fixation, bioabsorbable anchors could be a good alternative to metal anchors; the suture would eventually be integrated with the bone and would not have to rely on the anchor for fixation.

With either raised or embedded eyelets, another factor that likely affects suture failure is the sharpness of the eyelet edges. Here we define sharpness as the radius of a corner feature, where a lower radius corresponds to a sharper edge; previous finite element simulations have shown a strong relationship between cutting force and edge radius, though in a different context (blade cutting into soft material; *McCarthy, Annaidh & Gilchrist, 2010*; *Schuldt et al., 2016*). One study noted that, due to the small size of suture anchors, many have eyelets with sharp edges because their eyelets must be narrow and thin (*Meyer et al., 2002*), which may cause the suture to wear more rapidly. The suture anchor used in the rabbits had two edges that could be both considered sharp and contacted by the suture *in vivo*: the edges around the eyelet and the edge around the oval-shaped top of the screw (Fig. 3). In previous mechanical tests of suture anchors, edge radius or other potential aspects of sharpness (*e.g.*, wedge angle (*McCarthy, Annaidh & Gilchrist, 2010*) were not measured. Future studies should quantify the effect of edge sharpness on suture failure conditions, such as failure load, to inform the design of suture anchors.

Despite having a similar eyelet configuration as the Achilles tendons suture anchor but smaller suture, the tibialis cranialis suture anchors did not fail in any of the 12 rabbits. There are several factors that potentially explain the difference in failure rates between the suture anchors for the two different artificial tendons. For one, the Achilles tendon is expected to experience greater biomechanical loads than the tibialis cranialis tendon. The triceps surae-Achilles tendon unit provides body weight support against gravity and propels the body during locomotion (*Machado et al., 2021*); conversely, the tibialis cranialis muscle–tendon unit primarily contributes to foot motion during the swing phase of locomotion (*Viidik, 1969*). *In vivo* motion of the artificial tendon and suture relative to the anchor, which could affect wear of the suture against the anchor eyelet, could have been greater for the Achilles tendon than for the tibialis cranialis tendon. Finally, the eyelet design and material characteristics, including the radii (sharpness) and smoothness of edges, could have been more averse to suture durability for the Achilles tendon anchor, but this could be verified in future studies using optical profilometry.

The results of the Mann–Whitney U (non-parametric) and Fisher's exact tests provided further evidence to suggest that the suture breakage was caused in part by wear or abrasion of the suture against the anchor rather than exclusively due to biomechanical overload. These statistical tests found no effect of suture size on the timing or rate of failure. If overloading was the cause of failure, one may expect that failures would occur possibly later or at a lower rate for anchors with larger suture. This is especially true given that, as previously mentioned, the expected *in vivo* load ($57.7 \pm 0.5$ N, *West et al., 2004*) was substantially less than the strength of the largest suture used as reported by the manufacturer

(295 N, *Arthrex Inc, 2014*) and in our recent mechanical tests of the IMEX anchor (437 ± 42 N, *Fidelis et al., 2024*). Additionally, the non-parametric and Fisher's tests indicated no effect of the type of surgery (immediate *vs.* delayed replacement); this result suggests that the failure conditions were similar between the two surgery types.

In the present study, the survival analysis provided information about the likelihood of failure for the suture anchor used in the study along with their corresponding survival rates. The survival analysis of the suture anchors showed that only about 25 per cent of the sutures used to anchor the Achilles tendons survived up to 16 weeks post implantation, as verified by postmortem dissection of the limbs. These findings provide valuable insights into the durability and performance of the suture anchors over time, showing the proportion of sutures that effectively withstood the loading conditions in the rabbits post-implantation and for the duration of the study. By incorporating both tabular and graphical representations, this analysis offers a comprehensive understanding of the survival patterns and outcomes of the suture anchors under investigation. It will also assist in making important decisions about our choice of suture anchors for future research.

In order to reattach soft tissues, such tendons and ligaments, to bone, physicians and veterinary surgeons now frequently use suture anchors. These suture anchors are often used in veterinary surgery on the basis of clinician judgement rather than on the recommendations of the manufacturers. In humans and animals, as in our rabbit model, suture anchors are mostly expected to be loaded in a dynamic manner, with changes in both magnitude and orientation of load relative to the anchor. The results of this study shows that the anchors used for attaching the artificial Achilles tendon may not be suitable for this specific application, based on the multiple failures of the sutures of the suture anchors. This observation that could assist surgeons to make important decisions in both human and veterinary surgery so as to prevent costly revision surgeries which may be painful and burdensome to patients.

The reported suture anchor failures in the study are specific to artificial tendon implantation and may not be generalized to other applications of the suture anchor. The primary, traditional purpose of a suture anchor is to provide a stable attachment of a soft tissue (such as tendon) to a bone (Table S3). Attaching artificial tendons to bone using suture anchors is an emergent application and, thus, there are very few *in vivo* studies of this application. One previous study attached an artificial Achilles tendon to the calcaneus bone in rabbits using suture alone (*Melvin et al., 2003*). In the supplementary material, we present other methods used for bone-tendon attachment for previous polyester artificial tendon studies. Our results highlight some of the challenges in using one type of metallic screw-type anchor for artificial tendon attachment; future studies are needed to address these challenges or identify alternative, robust attachment methods.

This study has some limitations. The small sample size may have affected the outcome. Specifically, the study was not designed in such a way as to rigorously compare performance among suture anchors. Non-standard radiographic positioning may have affected gap measurement. The range of timepoints over which failure occurred (2–10 weeks post-surgery) as well as the standard deviation of the mean failure timepoint (standard deviation was 43% of the mean) were both relatively large. Additionally, the earliest failure timepoint

(2 weeks post-surgery) may seem too soon for wear to have occurred. However, previous *in vitro* mechanical tests observed that, for some suture anchors and loading conditions, the number of loading cycles to failure had similar variability and was low (<10 cycles) (*Bardana et al., 2003*). The radiographic evaluation time points were pre-planned based on the overall project goals and with the desire to reduce radiation exposure. Given the length of study, imaging every 2 weeks was considered appropriate. The current study of suture anchor failure was retrospective and, thus, we had limited control over the study variables. Variable initial gap length is a potential confounding factor that complicates our comparisons. Finally, there was no time-to-failure data upon which to plan more frequent imaging; this is an important consideration for future study planning.

## CONCLUSION

The suture anchor we used to attach artificial Achilles tendon to bone failed in 9 of 12 rabbits; no failures were observed for a different suture anchor used to attach tibialis cranialis artificial tendons. All failures of the Achilles tendon suture anchor were characterized by suture breakage at the mid-section (*i.e.,* away from the knot). Based on this failure mode and other aforementioned factors, we suspect that the failures were caused by wear of the suture against an edge of the suture anchor, which reduced the suture strength below *in vivo* loads. The failures may be partly attributed to the anchor design and to loading conditions. Therefore, the next steps in our study will include (1) measurement of suture anchor design features such as edge radii and (2) replication of the *in vivo* failures through laboratory testing of the strength of different suture anchors compatible with the rabbit model for attaching artificial tendons. The silicone coating may affect artificial tendon mechanical properties and future studies should investigate this effect.

## ACKNOWLEDGEMENTS

Thanks to the Office of Laboratory Animal Care staff for assisting with surgeries. Thanks to the animal housing facility staff for assisting with animal care. Thanks to Elizabeth Croy for setting up operating room and surgery supplies.

### Funding

This study was funded by the National Institutes of Health (R61AR078096) and the National Science Foundation (1944001). The funders had no role in study design, data collection and analysis, decision to publish, or preparation of the manuscript.

### Grant Disclosures

The following grant information was disclosed by the authors:
National Institutes of Health: R61AR078096.
National Science Foundation: 1944001.

## Competing Interests

The authors declare there are no competing interests.

## Author Contributions

- Obinna P. Fidelis analyzed the data, prepared figures and/or tables, authored or reviewed drafts of the article, and approved the final draft.
- Caleb Stubbs performed the experiments, prepared figures and/or tables, and approved the final draft.
- Katrina L. Easton performed the experiments, prepared figures and/or tables, authored or reviewed drafts of the article, and approved the final draft.
- Caroline Billings performed the experiments, prepared figures and/or tables, and approved the final draft.
- Alisha P. Pedersen performed the experiments, prepared figures and/or tables, and approved the final draft.
- David E. Anderson conceived and designed the experiments, performed the experiments, prepared figures and/or tables, authored or reviewed drafts of the article, and approved the final draft.
- Dustin L. Crouch conceived and designed the experiments, performed the experiments, prepared figures and/or tables, authored or reviewed drafts of the article, and approved the final draft.

## Animal Ethics

The following information was supplied relating to ethical approvals (i.e., approving body and any reference numbers):

University of Tennessee - Knoxville

## Data Availability

The data is available in the figures and tables.

## Supplemental Information

Supplemental information for this article can be found online at http://dx.doi.org/10.7717/peerj.18756#supplemental-information.

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
