# Peer review of "Attaching artificial Achilles and tibialis cranialis tendons to bone using suture anchors in a rabbit model: assessment of outcomes"

_PeerJ, doi:10.7717/peerj.18756_

## Round 0.1 · original submission · Major Revisions

Thank you for submitting your manuscript to PeerJ. After careful review, the reviewers have highlighted several important points that need to be addressed to strengthen the manuscript. We kindly request that you thoroughly address each of the reviewers' comments, providing detailed responses and justifications for any revisions or methodological choices.

Reviewer 1 ·

Basic reporting

The paper is written in clear, professional English, with suitable technical terminology for tendon repair and biomechanics. Nonetheless, the text should enhance its clarity in outlining the biomechanical distinctions between the tendons. The introduction offers adequate history on suture anchors in musculoskeletal reconstruction; however, it lacks references to more recent innovations, such as bioabsorbable anchors, and could be improved by additional illustrations of the suture anchor design used. The structure is systematically organized, featuring distinct parts for methodology, findings, and discussion, with figures and tables suitably labelled and pertinent. Providing mechanical testing data for the sutures would enhance clarity on the claims. The raw data is disseminated via a comprehensive analysis of outcomes. The work is self-sufficient and addresses its hypothesis; nevertheless, it could have been enhanced by incorporating mechanical testing of the sutures and addressing the material requirements particular to tendons.

Experimental design

The work presents a relevant and important research topic, focusing on the use of suture anchors for the anchoring of artificial tendons, thus solving a knowledge gap regarding tendon-bone attachment in animal models. Nonetheless, the study needs to more clearly elucidate how it tackles this knowledge gap, particularly through a comparative analysis of diverse suture materials and anchor designs (from prior work studies in a table). The experiment was conducted with rigorous ethical and technical standards, obtaining appropriate consent for animal procedures. The procedures are adequately specified for replication; however, the lack of pre-implantation mechanical testing and the rationale regarding the potential impact of varying suture materials on different tendons may limit the applicability of the findings.

Validity of the findings

The findings presented in the research are derived from robust and well-controlled data, accompanied by useful statistical analysis; nonetheless, the lack of pre-implantation mechanical testing weakens their validity. The paper, although not centered on novelty, effectively tackles a pertinent research gap in tendon-bone fixation, with conclusions suitably connected to the research question. The paper's limitations, especially the consistent application of suture materials across various tendons, necessitate a more comprehensive discussion in the conclusions. The incorporation of a silicone layer on the suture requires clearer explanation, as it may modify the suture's mechanical properties. This underscores the necessity for additional cyclic mechanical testing of both the suture and the silicone-coated type before the research. The findings are valid within the study's parameters; nonetheless, more research, particularly regarding material variation and pre-study testing, is recommended to enhance reproducibility and generalization.

Reviewer 2 ·

Basic reporting

The manuscript aims to identify the mechanisms underlying suture failure by implanting artificial tendons to rabbit models. It is clinically interesting that the authors tested the role of suture size and replacement timepoints in suture failure, but surprisingly neither of them affected the results.

Experimental design

In Introduction section, the authors discussed the mismatched (Line 96-97) mechanical properties of suture with these of Achilles tendon could cause the failure. However, mechanical properties of suture USP size #1, 2, and 5 used in the manuscript were not characterized or described. It is unknown whether the failure was mainly driven by suture strength itself instead of the factors checked here.
The authors evaluated different suture sizes, but suture size #2 only had one sample. The so small sample size made it challenging to find significant difference.
Figure 2 is very confusing. I am wondering why, for example, R1, R2, R7, and R12 as immediate tendon replacement were distributed along the timeline. 8 samples were used for immediate tendon replacement, but 4 samples were used for delayed tendon replacement. It is unclear why these two groups have different sample sizes.
It is helpful that gap length on the day of surgery was recorded. If gap length is smaller, high tension exists between anchor and suture so that they should be higher failure, as demonstrated by Table 1. Inconsistent gap length across samples complicated the comparison of different suture sizes and replacement timepoints.

Validity of the findings

There are so many factors which could impact the results, such as anchor design, suture types, suture sizes, gap length etc. Since the main focus of this study was suture sizes and treatment timepoint, other factors should be kept the same for better comparison. However, the manuscript discussed so many factors and even included Figure 3 to show this, which is very distracting for the readers.

Reviewer 3 ·

Basic reporting

see attachment

Language: The language is good and easy to understand
Introduction and literature review: Basically, the introduction is well written and summarizes international literature and the relevant topics to anchor fixation. Specific comments see below.
Structure is adequate and conforms to articles in the medical field.
Figures are of sufficient quality, except figure 1, which should be improved (shadows

Experimental design

see attachment
Raw data supplied is mostly sufficient, except pathological reports about the tendon rupture as it showed in postmortem evaluation is missing, resp. insufficient. Also mechanical data about load of tendons is presented without scientific data or literature reference.

Original primary research: the manuscript deals with a primary research study, although it is part of a larger study that is not reported here.
Research question well defined, relevant and meaningful. Fills an identified knowledge gap: the research question is not clearly defined. A hypothesis is missing.
Methods described with sufficient detail & information to replicate. Basically yes, for details see specific comments.

Validity of the findings

see attachment
Impact and validity: It is clear, that the manuscript as presented is part of a larger study dealing with the performance of an artificial tendon. The anchors and suture stability are a side effect, now reported in a separate paper. The impact of the artificial tendon on the stability of the sutures vs. fixation of natural tendons is not shown (no controls), and therefore at this time point it may be debated whether the holding power of the anchors and its sutures may be improved in natural tendons. It could be expected that fibrosis of the tendon and thus, holding power, may be better. Fibrosis and with this fixation around a silicon-coated tendon would be expected to be less than in a natural tendon without silicon sleeve which avoids adhesions (Meyer et al. J Orthop Res., 22(5):1004-7, s2004). Therefore, the conclusions must be restricted to the performance of these anchors in combination with the artificial tendons but cannot be extended for performance in general. This reduces the overall impact somewhat, especially in a manuscript standing alone.

Additional comments

See attachment
with specific comments to each section

Annotated reviews are not available for download in order to protect the identity of reviewers who chose to remain anonymous.

---

## Round 0.2 · accepted · Accept

I have assessed the revised version and can confirm that the authors have thoroughly addressed all the reviewers' concerns. The manuscript is now suitable for publication.